# ABCB1 Does Not Require the Side-Chain Hydrogen-Bond Donors Gln^347^, Gln^725^, Gln^990^ to Confer Cellular Resistance to the Anticancer Drug Taxol

**DOI:** 10.3390/ijms22168561

**Published:** 2021-08-09

**Authors:** Keerthana Sasitharan, Hamzah Asad Iqbal, Foteini Bifsa, Aleksandra Olszewska, Kenneth J. Linton

**Affiliations:** Blizard Institute, Barts and the London School of Medicine and Dentistry, Queen Mary University of London, 4 Newark St, London E1 2AT, UK; k.sasitharan@se17.qmul.ac.uk (K.S.); h.a.iqbal@se17.qmul.ac.uk (H.A.I.); f.bifsa@se17.qmul.ac.uk (F.B.); a.olszewska@se18.qmul.ac.uk (A.O.)

**Keywords:** P-glycoprotein, multidrug resistance, ABCB1, taxol, drug transport, ABC transporter

## Abstract

The multidrug efflux transporter ABCB1 is clinically important for drug absorption and distribution and can be a determinant of chemotherapy failure. Recent structure data shows that three glutamines donate hydrogen bonds to coordinate taxol in the drug binding pocket. This is consistent with earlier drug structure-activity relationships that implicated the importance of hydrogen bonds in drug recognition by ABCB1. By replacing the glutamines with alanines we have tested whether any, or all, of Gln^347^, Gln^725^, and Gln^990^ are important for the transport of three different drug classes. Flow cytometric transport assays show that Q347A and Q990A act synergistically to reduce transport of Calcein-AM, BODIPY-verapamil, and OREGON GREEN-taxol bisacetate but the magnitude of the effect was dependent on the test drug and no combination of mutations completely abrogated function. Surprisingly, Q725A mutants generally improved transport of Calcein-AM and BODIPY-verapamil, suggesting that engagement of the wild-type Gln^725^ in a hydrogen bond is inhibitory for the transport mechanism. To test transport of unmodified taxol, stable expression of Q347/725A and the triple mutant was engineered and shown to confer equivalent resistance to the drug as the wild-type transporter, further indicating that none of these potential hydrogen bonds between transporter and transport substrate are critical for the function of ABCB1. The implications of the data for plasticity of the drug binding pocket are discussed.

## 1. Introduction

Multidrug resistance (MDR) remains a problem for the chemotherapy of cancer patients [1,2]. Polyspecific efflux transporters of the plasma membrane that prevent the accumulation of a range of drugs to cytotoxic levels are a common cause of MDR [3,4]. The primary transporter associated with failure of chemotherapy is ABCB1 (previously known as P-glycoprotein and MDR1). Outside of the cancer clinic, ABCB1 is an important determinant of drug absorption, distribution, and excretion of many drugs including antibiotics, anti-epileptics and antiarrhythmics due to its native expression in a range of tissues including the apical membranes of gut epithelia and endothelial cells of the blood-brain barrier, and the canalicular membrane of the liver hepatocytes [5]. The polyspecificity of ABCB1 for many drugs of different structure and chemical class is a key feature of the transporter that needs to be understood to allow for drug designs which avoid recognition by the transporter and for the design of specific inhibitors. In this regard, the recent report by Alam et al. [6] of the structure of ABCB1 in complex with a transport substrate, the anticancer drug taxol, represents a milestone in its field. The complex was solved by single particle imaging using cryoelectron microscopy (cryoEM) with the taxol molecule occluded within the transmembrane domains of the transporter. Twelve amino acids were involved in drug coordination. These were primarily via weak Van der Waals interactions. However, three glutamines, Gln^347^, Gln^725^, and Gln^990^, located respectively in transmembrane helix (TMH) 6, TMH7, and TMH12, were highlighted to form hydrogen bond contacts (Figure 1). The study was not without limitations. The medium resolution of 3.5 Å, the low level of functional activity of the transporter within the nanodisc particles and the trapping of the transporter conformation using an inhibitory antibody may impact the physiological significance of the findings. The affect these have on the veracity of structural detail remain unclear and the implications of the binding pocket were not tested further.

Prior structure-activity studies, where “activity” refers to whether the one hundred chemicals that were tested are transport substrates of ABCB1 or not, had already suggested the importance of multiple free electron pairs (and their spatial pattern) in defining the transport substrates [7,8,9]. Taxol is particularly rich in these structural motifs with six pairs (or triplets) of acceptor sites separated by 2.5 or 4.6 Å, respectively, available for electrostatic interaction with the transporter. Two of these motifs were observed to form hydrogen bonds in the structure determined by Alam et al. (Gln^347^ and Gln^990^ coordinate the different oxygens within the same motif). Taken together, the simplest interpretation of these earlier structure-activity relationship data and the recent empirical structural data is that the three glutamines are likely key to drug recognition.

In the current study, we asked whether any or all of the glutamines are necessary for triggering the transport cycle and whether they have a role in the polyspecificity of the transporter. The glutamines were replaced with alanine residues to create single, paired and triple mutants. We then measured the effect on the transport efficiency of three different classes of drug in transiently-transfected cells and tested whether the mutants are able to confer resistance to taxol after stable expression in Flp-In cells.

## 2. Results

To test the importance of the hydrogen bond donors implicated in the coordination of taxol, we effectively removed the donor side chains by site directed mutagenesis and measured the function of the mutant protein in its native environment of live human cells. Function analysis used a modified flow cytometric transport assay to correlate ABCB1 expression with the reduced cellular accumulation of a fluorescent transport substrate normalizing to the untransfected cells in the population.

### 2.1. Endpoint Two-Colour Flow Cytometry Assay Measures the Function of ABCB1 in Live Cells

HEK293T cells were transfected transiently with pABCB1 encoding wild-type ABCB1. The density of ABCB1 on the cell surface was determined using saturating amounts of the anti-ABCB1 monoclonal antibody (4E3) that does not inhibit transporter function. The primary antibody was detected using saturating amounts of anti-mouse secondary antibody conjugated to a red fluorescent fluorophore. Use of saturating levels of the primary and secondary antibodies, which we determined previously [10], is important to confirm that the mutations introduced did not alter the expression level of the transporter in the plasma membrane. The red fluorescent cells are easily detected by flow cytometry and when they express the wild-type transporter these cells accumulate low levels of green-fluorescent transport substrates or drugs (Figure 2). Transient transfection under the conditions used never results in 100% transfection efficiency, so there are always non-expressing cells in the population and these become important internal controls. The drug content (green fluorescence) within the ABCB1-negative (untransfected) cells divided by the drug content of the ABCB1-positive cells (equation in Figure 2b) provides a robust and reproducible measure of the functionality of the transporter. The example given in Figure 2b shows that the wild-type transporter is able to maintain a fold difference of 96 over the untransfected population for Calcein-AM added to a final concentration of 0.5 μM. For comparison, for the Walker B double mutant which is unable to hydrolyse ATP and so cannot function as a primary active transporter, the calculated ratio is 1.3 (Figure 2c).

For these data to reflect the functionality of the expressed transporter in a reproducible manner it is important to control two factors. First, the cells must be exposed to a sufficient concentration of drugs such that all ABCB1 molecules are required to function to limit drug accumulation. This was determined empirically by exposing the cells to increasing levels of drug to determine the concentration at which the ABCB1-positive cells began to accumulate the green fluorophore. The titration of Calcein-AM is shown in Appendix A and shows that 0.5 μM is appropriate. Appendix A show the drug accumulation curves for drugs BODIPY-verapamil and OREGON GREEN taxol bisacetate (OG-taxol), respectively. Accumulation of all three drugs is non-saturable at least up to 5 µM for Calcein-AM, 8 µM for BODIPY-verapamil and 4 µM for OG-taxol consistent with a mechanism of entry by passive diffusion. The available data suggests that drug interaction with ABCB1 is directly from the lipid phase of the bilayer and is thus post desolvation of the drug [11]. Second, the experimental set up allows gating on populations of cells that express equivalent levels of the transporter (that have bound equivalent levels of the red fluorescent secondary antibody), as shown for the exemplar Calcein-AM transport experiment in Appendix A. This ensures that any differences in drug accumulation between mutants are due to the functionality of the transporter rather than the density of the transporters in the plasma membrane.

### 2.2. Gln^347^, Gln^725^ and Gln^990^ Modulate the Transport of the Xanthene Dye Calcein-AM

In order to test the importance of the hydrogen bonds formed between taxol and the side chains of ABCB1 amino acids Gln^347^, Gln^725^, and Gln^990^, the wild-type glutamines were replaced with alanine by site-directed mutagenesis. Alanine with its non-polar side chain is unable to donate an equivalent hydrogen bond to the transport substrate but its smaller side chain should be tolerated in the tertiary structure of the protein. Individual, pairwise mutants and a triple (Qtriple) mutant were generated.

#### 2.2.1. Of the Single Mutants, Only Q347A Shows a Statistically Significant Reduction in Calcein-AM Transport

Single mutants had only a modest effect on the transport of Calcein-AM as shown in the first four columns of Figure 3. Of the single mutants, only Q347A reduced the function of the transporter (to 63% of the wild-type level). Q725A showed a trend towards being more functional than the wild-type but this did not reach statistical significance and no effect was recorded for Q990A. This suggested that none of the hydrogen bonds that can be formed with the side chains of these three glutamines are essential for triggering the transport cycle although it is more efficient at effluxing Calcein-AM with Gln^347^ present.

#### 2.2.2. Q347A and Q990A Act Synergistically to Reduce the Transport of Calcein-AM

All three double mutant combinations and the triple mutant (Qtriple) were generated. Their ability to transport Calcein-AM showed pronounced and unexpected differences. A simple additive effect of Q347A and Q990A would predict a functionality of the Q347/990A double mutant to be 57% (the level of functionality of the Q990A mutant multiplied by the level of functionality of the Q347A mutant; 90/100 × 63/100 = 57/100). The observed activity of the Q347/990A mutant was reduced to 8.8% of the wild-type transporter suggesting a synergistic effect of the combined mutations and the importance of these two hydrogen bond donors for the efficient efflux of Calcein-AM. However, it is clearly not essential that these two glutamines are present because the 8.8 ± 1.87% (mean ± SEM) level of activity remains statistically higher than the Walker B mutant E556/1201Q (1.3 ± 0.59%) which is unable to hydrolyze ATP, indicating that the Q347/990A mutant retains measurable function.

#### 2.2.3. The Q725A Mutation Improves the Efficiency of Transport of Calcein-AM

The most surprising result is that the mutation of glutamine 725 to alanine improves the functionality of ABCB1 for the transport of Calcein-AM. This is most clearly evident when the Q725A mutation is introduced into the Q347/990A background to generate the Qtriple mutant. The Qtriple mutant has a transport activity of 50.7 ± 4.0% while the Q347/990A double mutant has a transport activity of 8.8 ±1.9%. The *p* value for this comparison by Student’s *t*-test is 0.0055, strongly suggesting that the inclusion of the Q725A mutation has made the Qtriple mutant more active than the Q347/990A double mutant and at the same time emphasizing that neither Gln^347^ nor Gln^990^ are absolutely necessary for ABCB1 to transport Calcein-AM. The ostensible increase in the mean activities of the other three constructs that include the Q725A mutation when compared to their respective backbones (wild-type versus Q725A, Q347A versus Q347/725A and Q990A versus Q725/990A) fail to reach statistical significance (Table A1).

### 2.3. Gln^347^, Gln^725^ and Gln^990^ Also Modulate the Transport of the Phenylalkylamine BODIPY-Verapamil

To test whether Gln^347^, Gln^725^, and Gln^990^ are also important for the transport of a different drug class, the transport assays were repeated with a fluorescent derivative of the phenylalkylamine verapamil, which acts as a calcium channel blocker and is used clinically to treat a variety of heart arrhythmias.

#### 2.3.1. Q725A Improves the Transport of BODIPY-Verapamil in Any Background

The challenge with BODIPY-verapamil gave a clearer indication that the side chain of the native Gln^725^ inhibits transport activity (Figure 4). Comparing the raw transport data of the Q725A single mutant with the wild-type transporter, it is statistically clear that Q725A increased the ability to efflux BODIPY-verapamil. This relationship is maintained for all mutants that include Q725A compared to the backbone into which the mutation was introduced. Thus, Q347/725A is statistically more active for the transport of BODIPY-verapamil compared to Q347A. Likewise, Q725/990A is more active than Q990A, and the Qtriple mutant is more active than Q347/Q990A.

#### 2.3.2. Q347A and Q990A Also Act Synergistically to Reduce the Transport of BODIPY-Verapamil

The Q347A and Q990A mutants (normalized transport activity of 81 ± 8% and 105 ± 16%, respectively (Appendix B Table A2)) combine in the Q347/990A double mutant to reduce the transport of BODIPY-verapamil to 38 ± 10% of the wild-type level (Figure 4). This double mutant also retains the ability to efflux BODIPY-verapamil because it is significantly different to the Walker B mutant E556/1201Q.

### 2.4. Gln^347^, Gln^725^ and Gln^990^ Have a More Limited Effect on the Transport of the Taxane Diterpenoid Derivative OG-Taxol

Taxol, the transport substrate that was first observed in complex with ABCB1, is not fluorescent but its derivative OREGON-GREEN taxol bisacetate (OG-taxol) fluoresces in the green spectrum and retains an ability to bind to microtubules in live cells. We tested whether, like taxol itself, it is also a transport substrate of ABCB1. A drug titration experiment showed that ABCB1-expressing cells accumulate less OG-taxol than non-expressing control cells and indicated that 0.4 μM OG-taxol was sufficient to require all of the ABCB1 molecules on the surface of our transiently-transfected HEK293T cells to limit accumulation of the drug (Appendix A).

#### 2.4.1. Of the Single Mutants Only Q990A Appears to Reduce the Transport of OG-Taxol

The Q347A and Q725A mutants were not distinguishable from the wild-type transport activity. However, the reduced activity of the Q990A mutant reaches statistical significance only after the raw data are paired (Figure 5). The effect is subtle with the Q990A mutant retaining 66 ± 8% transport activity for OG-taxol when normalized to wild-type ABCB1.

#### 2.4.2. The Double Mutants Q347/990A and Q725/990A Reduce the Transport Activity Further but the Triple Mutant Restores Wild-Type Levels of OG-Taxol Transport

The Q347/725A double mutant is trending towards reduced transport of OG-taxol but does not reach statistical significance. However, both Q347/990A and Q725/990A have reduced transport activity for OG-taxol, emphasizing the negative effect of the Q990A mutation. Perhaps surprisingly, given that all pairwise mutants seem to have reduced transport of OG-taxol, the triple mutant restores transport activity to wild-type levels.

#### 2.4.3. There Is No Indication That Gln^725^ Is Inhibitory for the Transport of OG-Taxol

In contrast to the transport of Calcein-AM and BODIPY-verapamil, there is no evidence from the data that transport of OG-taxol is improved in any mutant harboring the Q725A change. Although the Q347/990A mutant has a lower mean transport activity (48 ± 9%) than the Qtriple (82 ± 13%) these are not statistically different and none of the other mutants to which Q725A has been introduced come close to a statistically relevant difference to the parent plasmid (e.g., Q990A compared to Q725/990A).

### 2.5. The Q347/990A and the Qtriple Mutant Are Indistinguishable from the Wild-Type Transporter in Conferring Taxol Resistance to Cells in Culture

The more subtle differences observed for the transport of OG-taxol suggested that either the hydrogen bonds donated by Gln^347^, Gln^725^, and Gln^990^ were not particularly important for the transport cycle or that OG-taxol, despite being a transport substrate for ABCB1, does not replicate the geometry of taxol in the binding pocket. To test this, stable cell lines were derived to express the Q347/990A double mutant and the Qtriple mutant. The Q347/990A mutant was chosen because it consistently had the biggest effect on the transport of the three transport substrates tested and the Qtriple was chosen in case the three hydrogen bonds were critical only for the binding of taxol that they had been observed to coordinate. To ensure that like for like comparisons could be made, the cDNAs for Q347/990A and Qtriple were subcloned into pcDNA5/FRT. This allowed site-directed recombination to introduce a single copy of the plasmid into the same site within the genome of Flp-In HEK293. Along with Flp-In-ABCB1wt which was generated previously these stable cell lines ensured uniform levels of wild-type and mutant ABCB1 expression compared with the vector-only (integrated pcDNA5/FRT) negative control (Figure 6a).

The cell lines were challenged with increasing concentrations of taxol for three days, after which the cells with normal size and granularity were counted (Figure 6b; the use of a NovoCyte flow cytometer allowed all cells in the well to be counted in this experiment thus there is no estimation by counting only a small fraction of population). This allowed the IC_50_ for taxol to be calculated for the two test cell lines and compared to positive (cells expressing the wild-type transporter) and negative (cells with an integrated empty vector) controls (Figure 6c). It was clear from the survival curves that taxol is a potent cytotoxic, killing the vector-only control cells with an IC_50_ = 5.8 nM. Stable expression of wild-type ABCB1 shifts that IC_50_ more than 40-fold to an IC_50_ = 240 nM. The measured half maximal inhibitory concentrations for the Flp-In-Q347/990A and Flp-In-Qtriple cells are 430 nM and 480 nM, respectively, and are statistically indistinguishable from the effect of taxol on the Flp-In-ABCB1 wild-type cells.

## 3. Discussion

Structure-activity relationship (SAR) analyses [7] have indicated the importance of free electron pairs in the transport substrates of ABCB1 while structure data of the transporter in complex with taxol [6] has identified three glutamines within the drug binding pocket that donate hydrogen bonds to electron pairs in taxol. Molecular modelling studies [12] replicate these H-bond interactions in silico but their importance for drug recognition and to trigger the transport cycle remains unclear. Alam et al. [6] highlighted hydrogen bonds donated by Gln^347^, Gln^725^, and Gln^990^ to coordinate taxol in the binding pocket of ABCB1. We have tested whether this H-bonding pattern was key to triggering the transport cycle with three different pharmacophores, a xanthene, a phenylalkylamine, and two forms of a taxane diterpenoid. Several conclusions can be drawn from this study with the simplest being that none of these hydrogen bonds are absolutely essential for transport. Even the most impaired double mutant, Q347/990A, which retains only 8.8% of the wild-type level of activity for the transport of Calcein-AM, is still able to reduce the accumulation of the dye by cells in comparison to the non-functional Walker B mutant E556/1201Q. This equates to a 25-fold reduction in accumulation of Calcein-AM (the ratio of dye accumulation in the untransfected cells/Q347/990A-expressing cells) which is statistically different to the non-functional Walker B mutant which averages 1.3. The wild-type transporter, for comparison, can reduce accumulation of Calcein-AM by up to 258-fold in these experiments. It is thus clear that these mutant transporters which should lack the ability to donate hydrogen bonds to the transport substrate retain at least some level of transport activity for all three of the different classes of drug tested.

The situation, of course, is more nuanced. There is some consistency in the transport of different drugs. For example, the Q347/990A mutant has significantly reduced activity for the transport of all three fluorescent drugs but the level of impairment is to a different degree (8.8% of the transport activity of the wild-type for Calcein-AM, 37.8% for BODIPY-verapamil and 48% for OG-taxol). Thus, it would appear that the hydrogen bonding capacity of the side chains of Gln^347^ and Gln^990^ are involved in drug transport. There are also drug specific effects. There is a clear indication that introduction of the Q725A mutation improves the transport of Calcein-AM (cf. Q347/990A and Qtriple) and BODIPY-verapamil, but this is not true for OG-taxol. Perhaps the most surprising finding was that the Qtriple mutant, in which all three glutamines are replaced by alanine, retained activity (or regained activity compared to some of the double mutants) for the transport of all three drugs to achieve 51% transport activity for Calcein-AM, 116% activity for BODIPY-verapamil and 82% activity for OG-taxol. This observation also emphasizes that Gln^347^ and Gln^990^ are not critical for efficient transport because they are also absent from the Qtriple mutant which is indistinguishable from the wild-type transporter for the transport of BODIPY-verapamil and OG-taxol.

### 3.1. A Possible Allosteric Explanation for the Increased Transport Activity of Q725A

The negative effect of the wild-type Gln^725^ on the apparent transport activity is consistent with an earlier study by Loo et al. [13] during which they characterized a Q725C mutant (in an otherwise cysteine-less version of ABCB1). They observed that the basal ATPase activity of Q725C measured in vitro was raised 2.6-fold, offering a possible explanation for the improved transport of BODIPY-verapamil and Calcein-AM by the Q725A mutant if increased ATPase activity leads to increased drug efflux. It is possible that both the observed increase in ATPase activity and the increase in transport activity of fluorescent drugs when Gln^725^ is mutated is not due to the loss of an H-bond to the drug in the binding pocket but to the loss of an intra-molecular H-bond in a distinct conformation of the protein. In 2018, Kim and Chen reported the first medium resolution structure of human ABCB1 at 3.4 Ångstrom resolution [14]. They made use of the same E556/1201Q mutant used in the current study to prevent ATP hydrolysis and so were able to trap the protein with ATP bound in an ‘outward-facing’ conformation that is considered to show ABCB1 post drug release but prior to ATP hydrolysis. In this conformation, the binding cavity is closed to the membrane but open extracellularly. The side chains of Gln^347^ and Gln^990^ are not involved in electrostatic interactions with any other residue, but Gln^725^ forms a hydrogen bond with Asn^842^ of TMH9 (Figure 7). We speculate that the loss of this H-bond in the Q725A mutants may be more likely to increase the ATPase activity as TMH 9 is connected directly to the third ‘coupling helix’ (located at the base of the intracellular loop formed between TMH8 and TMH9). The coupling helices are thought to allosterically couple the drug binding pocket to the sites of ATP hydrolysis [15,16].

This leads us to an important caveat. The measurement of fluorescent drug accumulation by ABCB1-expressing cells compared to non-expressing cells is a robust test of ABCB1 function in live cells in its native environment of the plasma membrane, not just drug binding to the transporter. The rate of accumulation of drugs by cells depends also on the physicochemical properties of the drug to diffuse across the plasma membrane. The available evidence suggests that ABCB1 scans the membrane to identify and efflux hydrophobic compounds intercalated between the fatty acyl chains to preserve the chemical barrier [17]. The transport cycle begins when the drug complexes with the TMDs, triggering conformational change such that the NBDs bind ATP. The binding energy of ATP and the formation of the NBD:NBD interface is sufficient to change the conformation of the TMDs such that the drug binding site is reorientated to open extracellularly and affinity is lowered. Completion of the transport cycle requires ATP hydrolysis (the step that is absent from the E556/1201Q mutant) to drive the transporter back into the inward open conformation. Whilst our study of Gln^347^, Gln^725^, and Gln^990^ is predicated on the coordination of taxol it is also possible that the changes introduced could have a role to play as the transporter transitions through the conformational changes required to complete a full cycle.

### 3.2. Induced Fit of the TMDs around the Transport Substrate Likely Explains the Lack of Importance of Gln^347^, Gln^725^ and Gln^990^ in Conferring Resistance to Taxol

The flow cytometric assay is also limited by its dependence on fluorescent dyes and drug analogues. In OG-taxol bisacetate, while the oxygen atoms in the taxol pharmacophore that were observed to hydrogen bond with Gln^347^, Gln^725^, and Gln^990^ remain available, the molecular weight of the fluorescent drug is 1.5 times greater than that of taxol which must affect the geometry of the drug within the binding pocket. To rule out a possible artefact we generated stable cell lines and, to our surprise, demonstrated that the mutant Q347/990A and the Qtriple transporters conferred resistance to taxol to the same degree as the wild-type transporter. So, in a direct test against unmodified taxol that would be unable to hydrogen bond with the side chains of alanines at positions 347 and 990 (plus or minus position 725) we observed no difference in the survival curves of ABCB1-expressing cells, making it clear that the hydrogen bonds observed in the cryoEM data are not particularly important for the transport of taxol.

A second structure of ABCB1 in complex with a transport substrate, the vinca alkaloid vincristine, has now been reported by Nosol et al. [18]. The binding pocket of vincristine overlaps with that of taxol sharing six amino acids in common, including Gln^347^ and Gln^990^ (the latter considered close enough to hydrogen bond). Six further amino acids are unique to the vincristine pocket and five more for taxol. Some of these differences involve a subtle turn of a helix (for example Tyr^307^ in TMH5 is implicated in taxol binding while the adjacent amino acid Ile^306^ is implicated in vincristine binding). The main contributors to both binding pockets are from TMH5, 6, 11 and 12 while the vincristine pocket also includes Met^68^ and Met^69^ from TMH1 and Glu^875^ from TMH10, and the taxol pocket includes Gln^725^ from TMH7. It is perhaps not coincidental that Seelig had already noted in 1998 that TMH4, 5, 6, 11, and 12 are enriched in amino acids with hydrogen bond donor side chains [7]. It is possible to reconcile the drug SAR data, the empirical structural data, and the lack of effect of Q347/990A or the Qtriple to change the level of taxol resistance if the transmembrane domains are sufficiently flexible to fold around the transport substrate. An induced fit model has long been postulated to explain the unusually broad polyspecificity of ABCB1 and the first evidence in support of induced fit was reported by Clarke’s group in 2003 in which they showed a changing cross-linking pattern within the transmembrane domains in response to different drugs [19]. With this in mind, it seems perfectly reasonable to suggest that taxol might hydrogen bond to Gln^347^, Gln^725^, and Gln^990^, but in their absence different hydrogen bonds (or other electrostatic or weaker Van der Waals interactions) may be formed as the transmembrane domains close around the drug in the cavity. Further experiments will be required to test whether the lack of effect of the double or triple glutamine to alanine mutants are due to redundancy among the hydrogen bond donors within ABCB1. However, it is clear that neither Gln^347^, Gln^725^, nor Gln^990^ are essential for taxol efflux.

## 4. Materials and Methods

### 4.1. Site-Directed Mutagenesis

The cDNA encoding ABCB1 including a 12 histidine carboxy-terminal tag was described previously [20]. This cDNA was subcloned into pCI-neo to generate pCI-neo-ABCB1-12his (henceforth designated pABCB1). The coding sequence was modified by lightning site-directed mutagenesis (Agilent Technologies, Santa Clara, CA, USA) following the manufacturer’s recommendations except for generation of the Q347A mutant where a lower annealing temperature of 50 °C was required. The individual mutants for Q347A, Q725A, and Q990A were made first followed by the sequential addition of the second and third mutations. As a negative control for the transport experiments, the catalytic glutamates within NBD1 and NBD2 (Glu556 and Glu1201, respectively) were mutated to glutamines, thus preventing activation of the water for nucleophilic attack on the bound ATP. This mutant was generated previously [21]. Being unable to hydrolyze ATP, this mutant becomes trapped in the ATP-bound state [22]. Each cDNA was fully sequenced to ensure veracity.

Mutagenic oligonucleotides with the new alanine codons emboldened (Table 1).

### 4.2. Transient Expression of ABCB1

HEK293T cells were grown in DMEM high glucose (ThermoFisher Sci, Waltham, MA, USA) supplemented with 10% fetal bovine serum (FBS; ThermoFisher Sci, Waltham, MA, USA) at 37 °C in a humidified incubator with 5% CO_2_. For transient transfection 1.2 × 10^6^ cells were seeded onto a T25 culture flask. Twenty-four hours post seeding the cells were transfected with 10 μg plasmid DNA in complex with 15 μg linear polyethylenimine (Sigma-Aldrich, St. Louis, MO, USA). Cells were cultured for a further 48 h before harvesting with TrypLE (ThermoFisher Sci, Waltham, MA, USA) and quenching of the trypsin with culture medium.

### 4.3. Drug Transport Assay

Transiently-transfected HEK293T cells (5 × 10^5^) were incubated at 4 °C for 20 min with saturating levels (0.5 μg) of anti-ABCB1 antibody (4E3; Abcam, Cambridge, UK) in transport buffer (DMEM, high glucose (4.5 g/L), minus phenol red, supplemented with 1% FBS. The cells were pelleted at 500 G for 2 min and the supernatant discarded. The cells were washed once in transport buffer and resuspended in warm (37 °C) transport buffer containing saturating levels (2.5 μg) of goat anti-mouse secondary antibody, conjugated to recombinant phycoerythrin (RPE; Dako, Santa Clara, CA, USA) and one of three green-fluorescent drugs at a final concentration, unless otherwise stated, of 0.4 μM OREGON-GREEN Taxol bis-acetate (OG-taxol; Invitrogen, Waltham, MA, USA), 0.5 μM Calcein-AM (ThermoFisher Sci, Waltham, MA, USA), or 0.8 μM BODIPY-verapamil (Invitrogen, Waltham, MA, USA). The cells were incubated at 37 °C for 20 min before pelleting and washing as before. The cells were then resuspended in a 400 μL transport buffer and kept on ice until flow cytometry. Single fluorophore samples were also included to control for spectral spillover during flow cytometry. Two-color flow cytometry was performed on a LSRII (Becton Dickinson, Franklin Lakes, NJ, USA). Fluorescence data from 10,000 cells of normal size and granularity were acquired in CellQuest software (Becton Dickinson, Franklin Lakes, NJ, USA) and analyzed in Flowjo (Becton Dickinson, Franklin Lakes, NJ, USA).

### 4.4. Stable Expression of ABCB1 in HEK293 Flp-in Cells

HEK293 Flp-In cells with stable expression of ABCB1 wild-type were generated previously [10]. The cDNAs encoding ABCB1-Q347/990A and ABCB1-Q347/725/990A were excised from their parent pCI-neo plasmids using BamHI/NotI restriction endonuclease double digests and subcloned into the equivalent sites of pcDNA5/FRT (ThermoFisher Sci, Waltham, MA, USA). The resulting pcDNA5/FRT-ABCB1-Q347/990A and pcDNA5/FRT-ABCB1-Q347/725/990A (Qtriple) were used to co-transfect HEK293 Flp-In cells (ThermoFisher Sci, Waltham, MA, USA) along with the pOG44 (ThermoFisher Sci, Waltham, MA, USA) as a source of Flp recombinase as described by the manufacturer. Stable transfected cells (Flp-In ABCB1-Q347/990A, Flp-In ABCB1-Q347/725/990A (Qtriple) and pcDNA/FRT as a vector-only negative control) were selected with hygromycin (200 μg/mL) and, once uniform expression of the mutant ABCB1 were confirmed, maintained in hygromycin (100 μg/mL).

### 4.5. Taxol Survival Curve

Flp-In-ABCB1, Flp-In-ABCB1-Q347/990A, Flp-In ABCB1-Qtriple, and Flp-In-vector control cells (1 × 10^4^) were seeded into a 96 well dish in 100 μL of DMEM with high glucose and 10% FBS but without hygromycin and allowed to attach for several hours. Taxol (Cambridge Bioscience, Cambridge, UK) was added to a final concentration ranging from 0 nM to 10 μM and the cells cultured at 37 °C in a humidified incubator with 5% CO_2_. After 72 h the media was aspirated, the cells were detached with 30 μL TrypLE Express (ThermoFisher Sci, Waltham, MA, USA) which was quenched with 75 μL transport buffer and transferred to flow cytometry tubes. The entire population of cells of normal size and granularity, gated on the zero-drug condition, were counted in an ACEA NovoCyte flow cytometer (Agilent Technologies, Santa Clara, CA, USA). Cell number data were analyzed in Prism version 8 (GraphPad Software, San Diego, CA, USA). Curve fitting was achieved used non-linear regression.

## Figures and Tables

**Figure 1 ijms-22-08561-f001:**
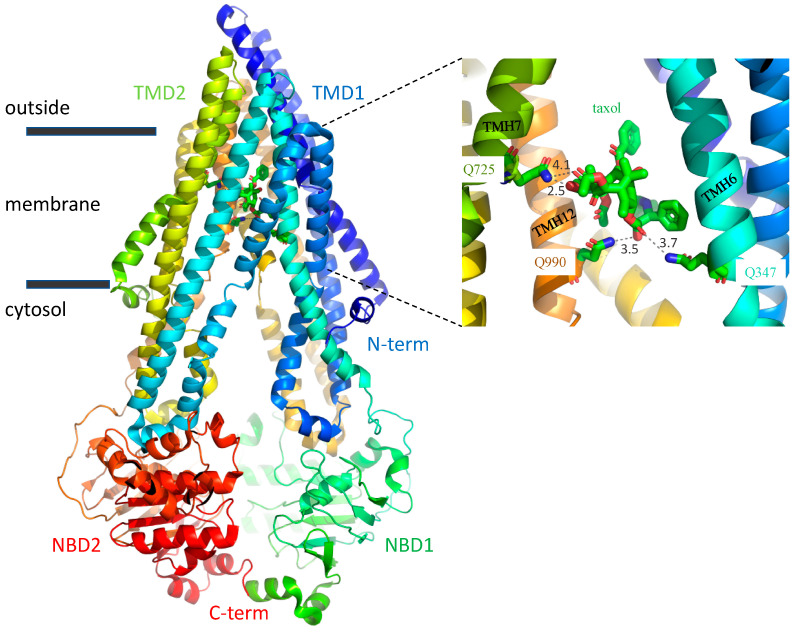
The taxol binding pocket. Ribbon depiction of ABCB1 with taxol occluded by the transmembrane domains (TMD1, blue-turquoise spectrum; TMD2, green-orange spectrum). The nucleotide binding domains, NBD1 and NBD2, are shown in green and red, respectively (pdb: 6QEX). The right-hand panel shows a 12Å slice in the Z plane of the taxol binding site. The three glutamines Gln^347^, Gln^725^, and Gln^990^ highlighted by Alam et al. [6] to hydrogen bond (dashed grey lines) with the taxol are show in single letter code and stick format with the bond lengths (N-O) indicated in black in Ångstroms. The combination of bond angle and length suggests that Gln^725^ forms the strongest and only H-bond with the baccatin III tetracyclic ring, while Gln^347^ and Gln^990^ form weaker H-bonds with the carbonyl and hydroxyl, respectively, which link to the diphenolic tail of the drug.

**Figure 2 ijms-22-08561-f002:**
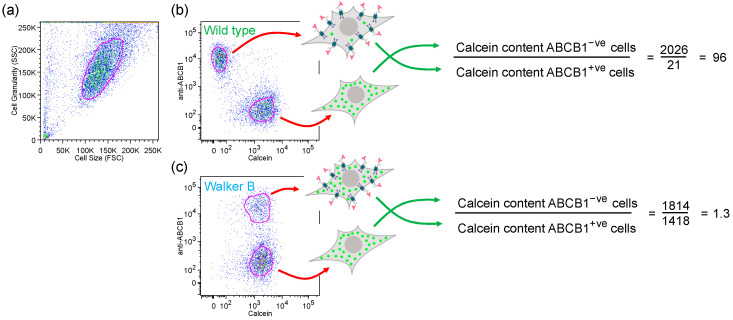
Measurement of functionality of ABCB1 for the transport of Calcein-AM by flow cytometry. (**a**) Dotplot showing the gating (autogated population circled in red) of HEK293T cells of normal size and granularity; (**b**) two-color dotplot of the gated normal cells showing green-fluorescent Calcein on the x-axis and red-fluorescent antibody binding on the y-axis. Cells that express wild-type ABCB1 bind the anti-ABCB1 antibody (red antibody shapes in the cartoon cell) and have a low level of accumulation of Calcein-AM (green circles in the cartoon cell) whilst untransfected cells accumulate high levels. The ratio of drug accumulation between the ABCB1-expressing and non-expressing cells is used to quantify transporter functionality. (**c**) Cells expressing the non-functional Walker B mutant (E556/1201Q) accumulate the same level of Calcein-AM as the untransfected cells in the population.

**Figure 3 ijms-22-08561-f003:**
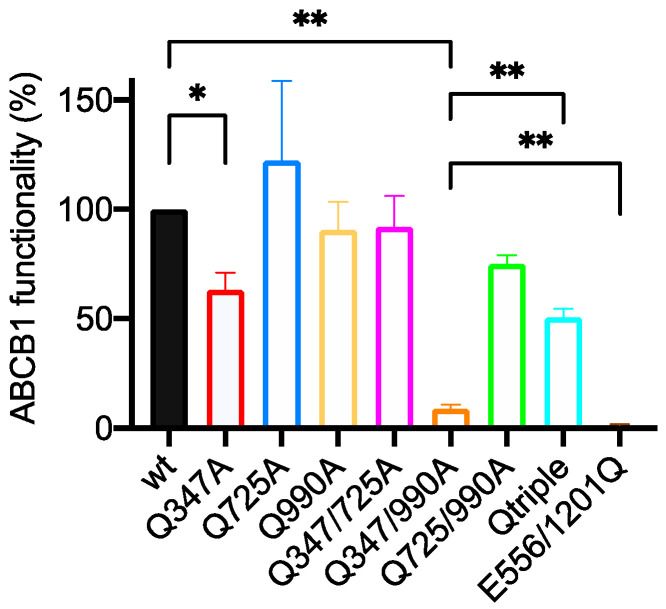
Functionality of glutamine to alanine mutants for the transport of Calcein-AM. Live HEK293T cells transiently expressing equivalent amounts of wild-type (wt) and mutant ABCB1 were challenged with Calcein-AM. Functionality was measured as the ratio of Calcein accumulation (Calcein-AM only becomes fluorescent once it is de-esterified in the cytosol) between the ABCB1-expressing and untransfected cells within the population. This was normalized to 100% for wild-type ABCB1 for the bar graph shown. The mean ± SEM was plotted using GraphPad Prism version 8; sample number was ≥3. Selected statistical analysis (ratio of paired Student’s *t*-test, two-tailed) performed on the raw data is shown with *p* values: * <0.05, ** <0.01. The full pairwise comparison of the data is given in Appendix B Table A1.

**Figure 4 ijms-22-08561-f004:**
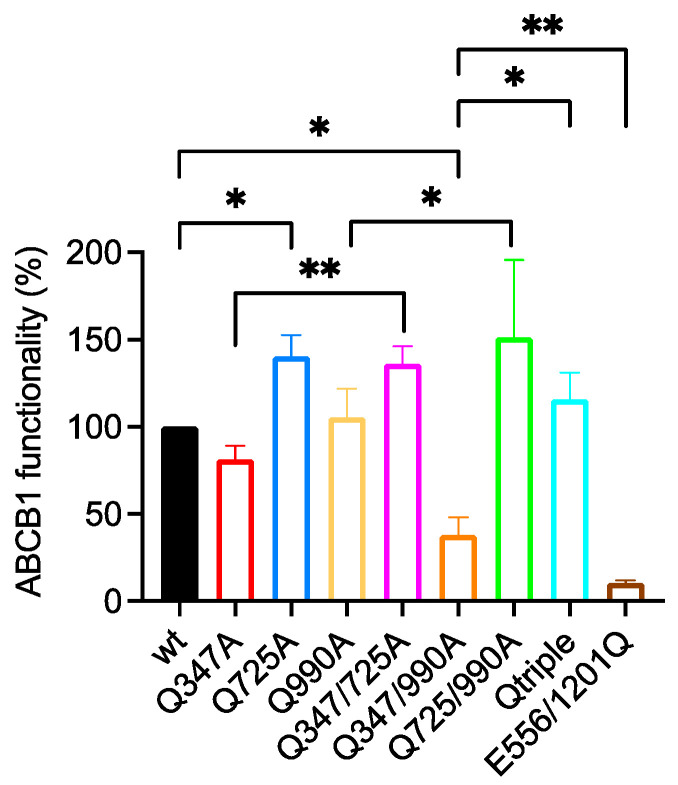
Functionality of glutamine to alanine mutants for the transport of BODIPY-verapamil. Live HEK293T cells transiently expressing equivalent amounts of wild-type (wt) and mutant ABCB1 were challenged with BODIPY-verapamil. Functionality was measured as the ratio of BODIPY-verapamil accumulation between the ABCB1-expressing and untransfected cells within the population. This was normalized to 100% for wild-type ABCB1 for the bar graph shown. The mean ± SEM was plotted using GraphPad Prism version 8; sample number was ≥3. Selected statistical analysis (ratio of paired Student’s *t*-test, two-tailed) performed on the raw data is shown with *p* values: * <0.05, ** <0.01. The full pairwise comparison of the data is given in Appendix B Table A2.

**Figure 5 ijms-22-08561-f005:**
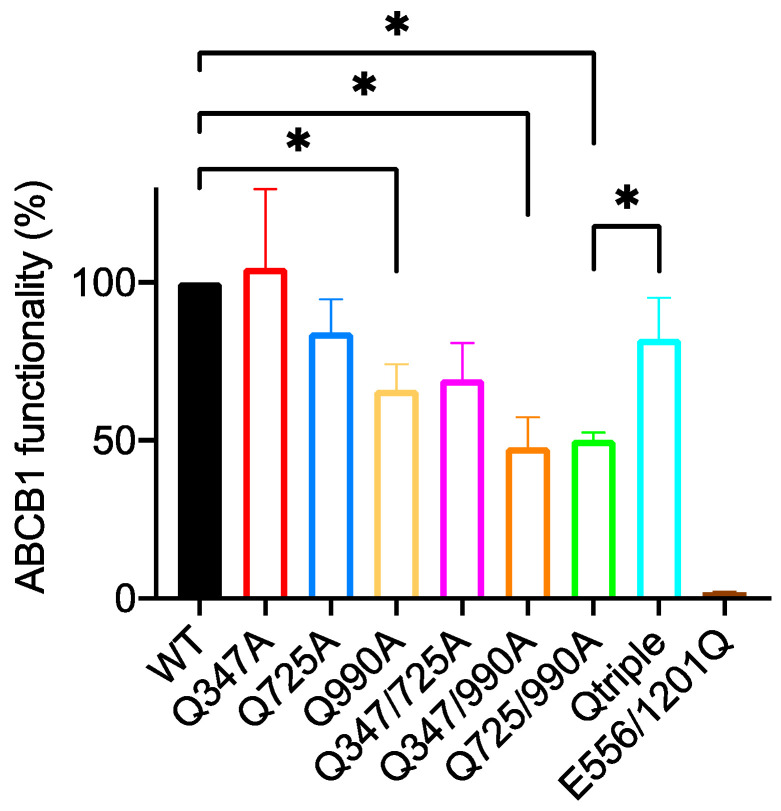
Functionality of glutamine to alanine mutants for the transport of OG-taxol. Live HEK293T cells transiently expressing equivalent amounts of wild-type (wt) and mutant ABCB1 were challenged with OG-taxol. Functionality was measured as the ratio of OG-taxol accumulation between the ABCB1-expressing and untransfected cells within the population. This was normalized to 100% for wild-type ABCB1 for the bar graph shown. The mean ± SEM was plotted using GraphPad Prism version 8; sample number was ≥3. Selected statistical analysis (unpaired Student’s *t*-test, two-tailed performed on the raw data except for the comparison of the wild-type with Q990A for which the raw data are paired) is shown with *p* value: * <0.05. The full pairwise comparison of the data is given in Appendix B Table A3.

**Figure 6 ijms-22-08561-f006:**
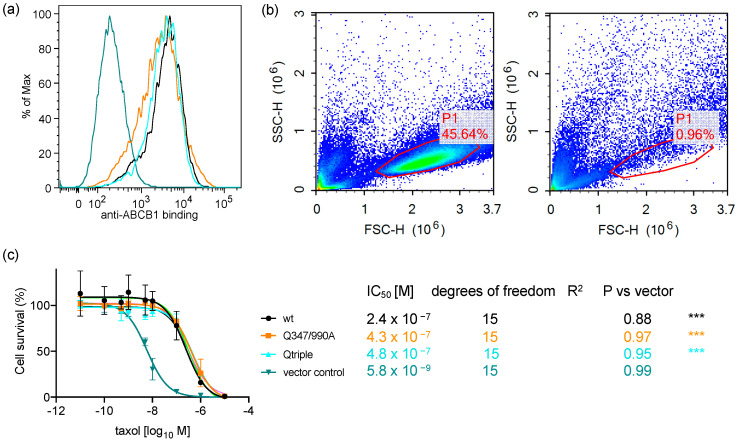
Stable expression of ABCB1-Q347/990A or ABCB1-Qtriple confers the same level of resistance to taxol as wild-type ABCB1. (**a**) Surface expression of ABCB1 (4E3 antibody binding) is similarly elevated for the two mutants (Q347/990A in orange and Qtriple in cyan) and the wild-type transporter (black) compared to the vector-only control (teal); (**b**) Flow cytometric exemplar dotplots in the presence and absence of taxol. Plots for the Flp-In ABCB1-Q347/990A cell line shows the forward scatter (FSC) and side scatter (SSC) heights in the absence (left hand plot) and the presence of 10 μM taxol for 72 h (right hand plot). All cells of normal size and granularity in the well were counted in a NovoCyte flow cytometer and analysed in NovoExpress software. The P1 gate which defines the healthy cell population in the absence of taxol was copied to all other conditions. In the examples shown, there were 71,969 cells in the P1 gate in the absence of taxol and 943 cells in the P1 gate following exposure to 10 μM taxol; (**c**) Non-linear regression analysis of cell survival on challenge with increasing concentration of taxol (colour code as above). Cell number in each well of the taxol dilution series was normalized to 100% for the P1 gate of the zero-taxol condition. The mean ± SEM was plotted with curve fitting by non-linear regression in GraphPad Prism version 8; sample number, *n* = 2 biological repeats. The biological repeats were averaged from duplicate technical repeats. *** *p* < 0.0001 compared to the vector-only cell line. The cell lines expressing the double and triple mutant ABCB1 are not significantly different to that expressing the wild-type transporter.

**Figure 7 ijms-22-08561-f007:**
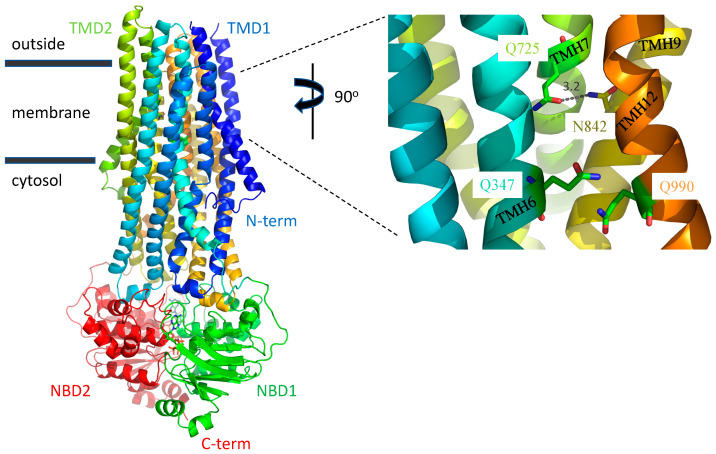
ABCB1 in the ATP-bound conformation focussing on the TMDs showing the positions of Gln^347^, Gln^725^ and Gln^990^ ‘post drug release’. Ribbon depiction of ABCB1 mutant E556/1201Q with ATP Mg^2+^ bound (pdb: 6C0V). The transmembrane domains (TMD1, blue-turquoise spectrum; TMD2, green-orange spectrum). The nucleotide binding domains, NBD1 and NBD2, are shown in green and red, respectively with two ATP molecules sandwiched at their shared interface depicted in stick format. The right-hand panel shows a 12 Å slice in the Z plane showing the positions of Gln^347^, Gln^725^, Gln^990^, and Asn^842^ in stick format and identified by single letter code. The hydrogen bond between Gln^725^ and Asn^842^ in this conformation is shown as a dashed grey line with the bond length (N-O) indicated in black in Ångstroms.

**Table 1 ijms-22-08561-t001:** Mutagenic oligonucleotides with the new alanine codons emboldened.

Q347A	
Forward	5′-ttaattggggcttttagtgttgga**gcg**gcatctccaagcat-3′
Reverse	5′-atgcttggagatgc**cgc**tccaacactaaaagccccaattaa-3′
**Q725A**	
Forward	5′-gtgccattataaatggaggcctg**gca**ccagcatttgcaataatatttt-3′
Reverse	5′-aaaatattattgcaaatgctgg**tgc**caggcctccatttataatggcac-3′
**Q990A**	
Forward	5′-gccatggccgtgggg**gca**gtcagttcatttgc-3′
Reverse	5′-gcaaatgaactgac**tgc**ccccacggccatggc-3′

## Data Availability

The raw data and materials generated in this study are available directly from K.J.L.

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
