# Peer review of "ABCB1 Does Not Require the Side-Chain Hydrogen-Bond Donors Gln347, Gln725, Gln990 to Confer Cellular Resistance to the Anticancer Drug Taxol"

_ijms, 2021, doi:10.3390/ijms22168561_

Round 1

Reviewer 1 Report

Sasitharan reported their experimental studies of mutagenesis on ABCB1 affect the transport of three different drug classes. The three residues, Q347, Q725 and Q990 were found to form hydrogen-bonds with taxol in a cryo-EM structure. The authors found that the single / double / triple mutations leading to very complicate patterns on the increasing/decreasing of the transport activities for those drugs. This work may be publishable if the authors can provide a rational/explanation/hypothesis on their findings.  The following is my suggestion for the authors to consider to further improve their manuscript:

  • The authors need to provide a picture showing the interactions between taxol and three GLN residues.
  • The authors need to make it clear in the text that the important role of a surrounding residue to ligand-binding is determined by many factors, including electrostatic interactions, van der Waals interactions, desolvation, and entropic effect, etc. Hydrogen-bond formation is a natural result of those interaction energies. Moreover, the importance of a HB strongly depends the distance between the donor and receptor and the bond angle of D-H…A. Thus, the three glutamine residues may not be the important residues at all.
  • Could the authors provide a rational or hypothesis to explain their experimental findings? What are the major factors determining the transport activity? What’s relationship between a drug’s transport activity and its binding affinity to the transporter?

Reviewer 2 Report

The paper titled “ABCB1 does not require the side-chain hydrogen-bond donors Q347, Q725, Q990 to confer cellular resistance to the anticancer drug taxol” reported the altered transportation of p-gp substrates by mutating several key amino acids. Specifically, when replacing glutamines at position 347 and 990 with alanines, the transportation function seemed to be reduced. Mutation at position 725 improved the transport of substrates. Overall, this is an interesting study. But the authors still need to address some issues before being considered to be published in IJMS.

The author studied three residues, Q347, Q725, Q990 and found they exhibited different alterations in the transport function of ABCB1. Specifically, Q347/990A showed synergistic effect on the transportation of dye-conjugated verapamil and calcein. And Q725A seemed to boost the function of ABCB1. But the mutants did not significantly change the sensitivity to taxol. The authors then concluded that, as indicated in the title, the three residues were not required in ABCB1 to confer taxol resistance. However, this conclusion is not fully supported by their results because they only tested the cell survival of Q347/990A. The data for all single and double mutant are necessary.

Moreover, some protein expression quantitative or semi-quantitative analysis is needed to show the expression level of different types of mutants.

Besides, there are some other issues:

  1. The authors need to provide more information regarding the control E556/1201Q (Walker B mutant).
  2. The data in Fig S1 does not have error bar. The authors need to provide data for repeated experiments to support their conclusion.
  3. In figure 3, the error bar of Q725/990A is almost 30%, the authors need more repeats to reduce the error bar and get meaningful results.
  4. The red text in Figure 5b (right panel) is not readable.
  5. In Figure 5c, the author should also provide data for the control E556/1201Q mutant
  6. The statement in line 303 – 304 “Q347/990A which has only 8.8 % of the wild-type level of activity for the transport of Calcein-AM, can significantly reduce the accumulation of the dye by cells.” Is not correct. Based on the results and conclusion, the double mutant should increase the accumulation since it decreased the ABCB1 function. In the following sentence, it is not accurate to say “reduction in uptake”. The authors did not provide any evidence to demonstrate reduced uptake. Should be “increased efflux” or similar statement, based on the function of ABCB1.
  7. In discussion, the author mentioned “This negative effect of Q725 on transport activities …”. But based on the results, Q725 at least showed positive effect on ABCB1 functions
  8. In table A1 and A2, the authors mentioned E556/1201A in the table legends. In text, E556/1201Q was mentioned a lot. This is confusing.

Round 2

Reviewer 2 Report

The authors have addressed most of my concerns. And the manuscript is properly improved.